# The Viromes of Mosquitoes from the Natural Landscapes of Western Siberia

**DOI:** 10.3390/v15091896

**Published:** 2023-09-08

**Authors:** Vladimir A. Ternovoi, Alexander N. Shvalov, Mikhail Yu. Kartashov, Eugenia P. Ponomareva, Natalia L. Tupota, Yuri A. Khoroshavin, Roman B. Bayandin, Anastasia V. Gladysheva, Tamara P. Mikryukova, Tatyana V. Tregubchak, Valery B. Loktev

**Affiliations:** 1State Research Center Virology and Biotechnology “Vector”, Federal Service for Surveillance on Consumer Rights Protection and Human Wellbeing of Russia, World-Class Genomic Research Center for Biological Safety and Technological Independence, 630559 Koltsovo, Russiakartashov_myu@vector.nsc.ru (M.Y.K.); ponomareva_ep@vector.nsc.ru (E.P.P.); tupota_nl@vector.nsc.ru (N.L.T.); horoshavin_yua@vector.nsc.ru (Y.A.K.); bayandin_rb@vector.nsc.ru (R.B.B.); gladysheva_av@vector.nsc.ru (A.V.G.); mikryukova_tp@vector.nsc.ru (T.P.M.); tregubchak_tv@vector.nsc.ru (T.V.T.); 2Department of Physics, Novosibirsk State University, 630090 Novosibirsk, Russia; 3Institute of Cytology and Genetics, 630090 Novosibirsk, Russia

**Keywords:** metagenomics, virome, viruses, mosquito, Novosibirsk region, western Siberia

## Abstract

The metagenomic analysis of mosquitoes allows for the genetic characterization of mosquito-associated viruses in different regions of the world. This study applied a metagenomic approach to identify novel viral sequences in seven species of mosquitoes collected from the Novosibirsk region of western Siberia. Using NGS sequencing, we identified 15 coding-complete viral polyproteins (genomes) and 15 viral-like partial sequences in mosquitoes. The complete sequences for novel viruses or the partial sequences of capsid proteins, hypothetical viral proteins, and RdRps were used to identify their taxonomy. The novel viral sequences were classified within the orders *Tymovirales* and *Picornavirales* and the families *Partitiviridae*, *Totiviridae*, *Tombusviridae*, *Iflaviridae*, *Nodaviridae*, *Permutotetraviridae*, and *Solemoviridae*, with several attributed to four unclassified RNA viruses. Interestingly, the novel putative viruses and viral sequences were mainly associated with the mosquito *Coquillettidia richardii*. This study aimed to increase our understanding of the viral diversity in mosquitoes found in the natural habitats of Siberia, which is characterized by very long, snowy, and cold winters.

## 1. Introduction

Mosquitoes are well known to transmit numerous arboviruses that cause viral infections in animals and humans, such as the West Nile virus (WNV) and the Zika, Japanese encephalitis, Chikungunya, and dengue viruses [1,2,3,4]. In recent years, the development of metagenomic approaches has led to the discovery of many novel viruses in invertebrates [5,6,7,8,9,10]. Studying the viromes of different species of mosquitoes has revealed new viruses referred to as insect-specific viruses (ISVs). The viral interference within the ISV group and pathogenic viruses may dramatically change the viral biodiversity in mosquitoes and, thereby, predetermine the transmission of pathogenic viruses from mosquitoes to animals and humans [11,12,13,14,15,16]. These are viruses belonging to the families *Peribunyaviridae*, *Flaviviridae*, *Reoviridae*, and *Togaviridae*, and each is a potential source of viral biodiversity for viruses, with dual tropism for invertebrate and vertebrate hosts [7,17,18].

Comparing the viromes of mosquitoes from different geographical regions has revealed their biodiversity, providing new insights into the phylogeography of mosquito-borne viruses [5,6,19,20]. Generally, such studies are conducted in countries characterized by warm or tropical climates, such as China, Australia, Mozambique, and the USA, where mosquito-borne viral infections are not uncommon. The Sindbis, Inco, and West Nile viruses are usually detected and isolated from mosquitoes in the southern regions of Russia [21,22]. No systemic information is available for western Siberia, where it is also possible for mosquito-borne viruses to circulate. This region has a continental climate with long winters and short summers that may limit the biodiversity of mosquito species and mosquito-associated viruses. Only a few studies have reported the detection of WNV markers in bird and human cases of West Nile fever in western Siberia [22,23]. 

In this study, we sought to investigate the biodiversity of mosquito-borne viruses from different mosquito species collected in western Siberia using metagenomic approaches.

## 2. Materials and Methods

### 2.1. Mosquito Samples

For this study, 3910 mosquitoes were collected in the Novosibirsk region during the spring–summer periods of 2017–2018. The collection sites were situated in typical mosquito habitats in western Siberia (Figure 1): deciduous and mixed forests with well-developed grassy cover, deforestations with the natural resumption of hardwoods, and the banks of streams. The mosquitoes were collected using a light trap (https://survinat.ru/2011/09/metodika-sborov-xraneniya-i-izucheniya-komarov/, accessed on 23 June 2023). Capture was conducted after sunset. The mosquitoes were transported in a thermal bag on a damp napkin at a temperature of 4 °C and stored at minus 18–24 °C. Fragments of the 16S rRNA and the COI gene of the mitogenome were sequenced via the Sanger method to determine the mosquito species [11]. Pools of 10–40 mosquitoes were formed according to the morphological features of mosquito species, and fragments of the 16S rRNA and the COI gene from individual mosquitoes were selectively sequenced to determine their species.

### 2.2. Sample Preparation

All the mosquitoes were washed with 70% ethanol and then rinsed twice with water, followed by homogenization to remove potential surface microorganisms. The homogenization of the samples was performed mechanically by grinding them in a mortar with 300 µL of sterile saline. The homogenates were centrifuged at 8000× *g* for 5 min at 4 °C, and the supernatants were used for the analysis. The total RNA was extracted using an Extract RNA reagent (Eurogen, Moscow, Russia) according to the manufacturer’s protocol and purified on Cleanup Mini spin columns (Eurogen, Moscow, Russia). The pools were then processed using benzonaze [24]. The first cDNA chain was synthesized using the NEBNext Ultra Direction module. The second cDNA chain was synthesized using the UMI Second Strand Synthesis module (Illumina, San Diego, CA, USA, Lexogen, Wien, Austria).

### 2.3. NGS Sequencing and Phylogenetic Analysis

The dsDNA libraries were prepared and analyzed via NGS on MiSeq using Illumina technology. Cutadapt (version 1.18) and SAMtools (version 0.1.18) were used to remove the Illumina adapters and re-read. The contigs were assembled de novo using MIRA assembly (version 4.9.6). The experimentally determined sequences were deposited in GenBank. The phylogenetic analysis was performed using RNA-dependent RNA polymerase (RdRp) sequences from GenBank with an amino acid identity > 20%. The sequences were aligned, and phylogenetic trees were built in Vector NTI Advance 11, MEGA 7/10 (PSU, USA), and Lasergen 7 (Invitrogen, Waltham, MA, USA). The resulting viral sequences and sequence read archive (SRA) were deposited in GenBank. The phylogenetic trees were calculated via the maximum likelihood method, using 500 replicates for bootstrap values.

## 3. Results

### 3.1. Mosquito Species

A total of 3910 mosquitoes were collected in the Novosibirsk suburbs and the rural district of the Novosibirsk region in 2017–2018. Pools of 10–40 mosquitoes were formed to identify the mosquito species for each collection point. Fragments of the 16S rRNA and the COI gene of the mitogenome were sequenced to determine the species of mosquitoes (Figure 2). *Aedes caspius* (Pallas, 1771) and *Ae. mariae* (Sergent and Sergent, 1903) were found to be the most abundant species, comprising 41.2% of the total. *Anopheles messeae* (Falleroni, 1926) accounted for 34.6%, *Culex pipiens* (Linnaeus, 1758) for 8.3%, *Coquillettidia richardii* (Ficalbi, 1889) for 5.5%, *C. modestus* (Ficalbi, 1889) for 4.8%, and *An. maculipennis* (Meigen, 1818) and *An. sinensis* (Wiedemann, 1828) for 2.8% each. 

### 3.2. NGS Sequencing

In total, 144 putative viral sequences were selected in the first step, and 30 of these sequences with lengths greater than 1239 bp were chosen. Eight sequences with a level of identity > 80% aa were previously described as mosquito-borne viruses (Table 1). These were Partitivirus-like 1 (dsRNA, *Partitiviridae*), Hammarskog tombus-like virus (ssRNA (+), *Tombusviridae*), Hammarskog picorna-like virus (ssRNA (+), *Picornavirales*, unclassified), Lymantria dispar iflavirus 1 (ssRNA (+) virus, *Iflavirus*), Wenzhou noda-like virus 6 (ssRNA (+) virus, unclassified), Mayapan virus (ssRNA (+), two segments), Sanxia permutotetra-like virus 1 (ssRNA (+) virus, unclassified), and Chaq virus-like 1 (RNA virus, unclassified). An additional 22 viral sequences were presented as putative mosquito-borne sequences, with a level of identity less than 79% for viral prototype sequences. 

The proportions for classified and unclassified viral reads are presented for 13 mosquito pools of Cq. richardii (Figure 3). The prevalence of unclassified and classified picornaviruses was detected in practically all the pools studied. Unclassified viral sequences were also analyzed, ranging from 0.13% to 34.27% for the different pools.

The phylogenetic analysis results for the sequences obtained from the mosquitoes are presented in Figure 4. This phylogeny was based on the amino acid sequences of RdRps, with these data confirming the biodiversity of the mosquito viruses found in nature. 

#### 3.2.1. Tymovirales (Positive ssRNA)

The complete viral genome of a tymovirus-like sequence with an identity level of 60% (according to its amino acid sequence) with a previously described insect-associated tymovirus 1 in Mexico (MN203215) was detected in the *Cq. richardii* mosquitoes. This virus has been designated as an Inya insect-associated virus 1 (MW251313; MW251314). The genomic positive ssRNA of the Inya insect-associated virus was identified as comprising 6526 bp and three ORF-encoding proteins (Appendix A). The ORF MP of Inya insect-associated virus 1 was an RdRp, and this ORF contained a highly conservative “tymobox” near the 5′-end [25]. The tymobox sequence had 16 nucleotides that were likely part of the subgenomic promoter for the third ORF encoding the coat protein (CP). Previously, the *Tymovirales* were well-known as plant viruses [26]. Inya insect-associated virus 1 can presumably be identified taxonomically as belonging to the order *Tymovirales*, unclassified *Tymovirales*.

#### 3.2.2. Partitiviridae (dsRNA)

Partitivirus-like 1 was detected in *Cq. richardii* and demonstrated an identity level of 89% with an isolate from *Anopheles gambiae* that was collected in Liberia (KX148575). Another novel putative partitivirus was detected in the pool of *Cq. richardii* mosquitoes (Appendix A) and was designated as Krahall insect-associated virus 1 and 2, with an identity level of 60–63% shared with a previously described Atrato partiti-like virus 2, which was isolated earlier from *Anopheles darlingi* in Colombia (MN661058). In addition, seven suspected partitiviruses were found in the pool of *Cq. richardii* mosquitoes. These are the novel insect Talaya 1 and 2 viruses (MW251327–MW251330), with aa sequence identity levels of 68–71% identity with a previously described Partitivirus-like 1 (Liberia, KX148575); the Tarbrook virus (MW251325), with 79% homology with a previously described Sonnbo virus in Sweden (MK440649); and Zeyabrook partiti-like_viruses 1 and 2, with similarities of 38–42% with a Beihai partiti-like virus 2 (NC_032500) from China. All the prototype sequences were isolated earlier from invertebrates (mollusks, octopuses, mosquitoes, and odonatos). These partitiviruses were preliminarily taxonomically identified as belonging to the family *Partitiviridae*, unclassified *Partitiviridae*.

#### 3.2.3. Totiviridae (Monopartite dsRNA)

We have found a novel putative totivirus, designated Zyryana toti-like virus 2, with the prototype Fitzroy Crossing toti-like virus 1 isolated earlier from *Culex annulirostris* in Australia (MT498830) (Table 1). The Zyryana toti-like virus 2, with an identity level of 51% with the prototype sequence, was detected in the pools of *An. Messeae* and *Cq. richardii* mosquitoes. The length of the nucleotide sequence of the Zyryana toti-like virus 2 was over 5863 bp, and the ORFs encoded two proteins, 1112 aa and 807 aa, of the CP and RdRp. The putative genome organization schemes for these totiviruses are presented in Appendix A. 

#### 3.2.4. Tombusviridae (Positive ssRNA)

Tombusviridae (Tolivirales, Tombusviridae) are single-stranded RNA (+) genomes between 3.7 and 4.8 kb in length, currently regarded as plant viruses with a relatively limited selection of hosts, and they usually require a subviral RNA for replication [27,28,29]. These viruses are frequently found in insects, and the results show that tombusviruses were first detected in mosquitoes in western Siberia. We present the complete polyprotein (4317 bp) for the Hammarskog tombus-like virus, which had 90% identity with a similar virus that was detected in Sweden (MN513379) and isolated from *Cq. Richiardii* in 2017 [30]. The 4166 bp partial polyprotein contained three ORF-encoded hypothetical polypeptides, 397 aa, 482 aa, and 409 aa, which differed for Hubei tombus-like 20 (Appendix A). In addition, we found a novel Oyosh tombus-like virus with 62% identity with Hubei tombus-like virus 13 (NC033017), which was isolated from house centipedes in China. Four polypeptides were encoded by a prototype genome (5904 bp). The RdRp for *Tombusviridae* was translated using a potential alternative mechanism to suppress the stop-codon-reading mechanism with the formation of a full-size protein with an elongated ORF1 C-end [6]. 

#### 3.2.5. Picornavirales (Positive ssRNA)

Most members of the order Picornavirales have a single molecule of positive-sense RNA ranging in length between 7000 and 12,500 nt. The viral RNA is infectious and serves as a template for replication and mRNA [31]. Six different picorna-like viruses were identified as mosquito-associated viruses in western Siberia (Table 1). We assembled a complete genome for the Hammarskog picorna-like virus (11,507 bp) from the *Cq. richardii* mosquitoes that had five OFRs encoding 175 aa, 156 aa, 121aa, 376 aa, and 2424 aa polypeptides with 98% aa identity with a previously described Hammarskog picorna-like virus (MN513381) isolated from *Cq. richiardii* in Sweden (Appendix A). In addition, other novel picorno-like viruses were found in the *Cq. richardii* mosquitoes collected in the Novosibirsk region. These are Miltyush picorna-like viruses 1 and 2, the Isses picorna-like virus, the Ichacreek insect virus, and the polycipiviridae associated with Ora rivulet insects. 

The Miltyush picorna-like virus was found to have only 36% identity with a previously detected Halhan virus 3 from Haliotis discus hannai in Korea (NC040628). Isses picorna-like virus 1 was found to have 64% identity with a Washington bat picornavirus previously discovered in the USA (KX580885). Ichacreek insect virus 3 was identified to have a 44% level of identity with a previously detected Solenopsis invicta virus 3 (GU017972) from *Solenopsis invicta* in Argentina. Ora rivulet insect-associated polycipiviridae were identified to have a 34% identity level with previously discovered *Polycipivirida* sp. isolated from *Pteropus lylei* in Cambodia (MK161350). The preliminary taxonomic identification for these six viruses is *Picornavirales*, unclassified *Picornavirales*.

#### 3.2.6. Iflaviridae (Positive ssRNA)

The order *Picornavirales* also includes some iflaviruses that were found in *Cq. richardii* mosquitoes in this study. Lymantria dispar iflavirus 1 was detected, with its sequence having 99% identity with already-known viral isolates from the USA and Russia (KJ629170, MN938851). The alignment and phylogenetic analysis revealed a high level of sequence identity with the representatives of *Iflavirus*, the family *Iflaviridae*.

#### 3.2.7. Nodaviridae (Bi-Partite Positive-Sense, ssRNA, Sometimes Associated with Subviral RNA Molecule Capable of Replicating Itself)

We assembled practically the whole genome for Wenzhou noda-like virus 6 from the *Cq. richardii* mosquitoes, as well as the Mayapan virus, with identity levels of 87% and 90%, respectively. Previously, the Wenzhou noda-like virus 6 sequence was identified in *Channeled applesnail* in China (KX883260), and the Mayapan virus (MH719096) was isolated from the *Psorophora ferox* mosquito in Mexico (Appendix A). Other novel nodaviruses were also found in the *Cq. richardii* mosquitoes collected in the Novosibirsk region. These were Mayzas noda-like virus RNA 2, with a prototype Mayapan virus RNA2 segment (MH719097), which had a 43% identity with that isolated from the *Psorophora ferox* mosquito in Mexico, and the insect-associated Uzakla virus with a prototype Mosinovirus (KJ632942), which had a 36% identity with that isolated from *Culicidae* spp. in the Cote d’Ivoire. 

#### 3.2.8. Permutotetraviridae (dsRNA)

The genomic RNA for permutotetraviruses is 4582 bp long and encodes three ORFs overlapping in a short region (Appendix A). The longest ORF (1028 aa) encoding the RdRp overlaps for 106 nucleotides with a small ORF (199 aa), which presumably encodes the capsid protein. Like all permutotetraviruses, the sequences from the *Cq. richardii* mosquito pools showed the presence of the virus with a 93% identity with Sanxia permutotetra-like virus 1, which was previously detected in water striders in China (KX883450). The Uzakla mosquito-associated permutotetra-like virus, with a 53% identity with Vespa velutina permutotetra-like virus 2 in France (MN5650551; MN565052), was described early on as belonging to the group of unclassified *Permutotetraviridae*.

#### 3.2.9. Other Viruses and Viral Sequences

Two variants of Chaq virus-like 1, Tartas insect associate virus, ZeyaBrook chaq-like virus 2, Kamenka insect-associated virus, and the Uzakla insect virus were detected in *Cq. richardii*. The Chaq virus-like 1 had 82% identity with an earlier described unclassified sequence from *Anopheles gambiae* in Liberia (KX148554). The ZeyaBrook chaq-like virus 2, Kamenka insect-associated virus, and Uzakla insect virus had identity levels of 33–47% with previously unclassified putative viral sequences (KX148556, KX883594, and NC032218, respectively) isolated from invertebrates. Only the Tartas insect associate virus may be classified as unclassified Solemoviridae, having an identity level of 61% with prototype Atrato Sobemo-like virus 6 (MN661101), which was detected in *Wyeomyia* spp. mosquitoes in Colombia. The *Solemoviridae* have a relatively small (4–4.6 kb) positive-sense, single-stranded, monopartite RNA genome with 4–5 ORFs, and they are usually associated with plant viruses. 

## 4. Discussion

The application of the metagenomic approach offers novel opportunities for virome analysis [5,7,20]. This approach has provided new insights into the evolution of viruses of clinical importance and has allowed new viruses to be discovered from different viral families, such as *Peribunyaviridae* [5], *Rhabdoviridae* [32], *Orthomyxoviridae* [7,33], *Flaviviridae* [34], and *Reoviridae* [35], as well as the unclassified *Chuvirus* [7] and *Negevirus* [36]. Recent metagenomic studies have also confirmed the presence of the dengue virus, Zika virus, and Japanese encephalitis virus in mosquitoes in China [37,38]. 

Numerous genetically diverse viruses have also been detected via NGS sequencing in plants, invertebrates, and vertebrates in tropical countries [7,8]. Phylogenetic analyses have demonstrated that it is possible for all host species and viruses to co-evolve by changing hosts. Mosquitoes are among the most common and important viral vectors of the Zika, dengue, yellow fever, and West Nile viruses and are associated with unprecedented global outbreaks of these infectious diseases in tropical countries [39,40]. In addition, mosquitoes are also known to carry insect-specific viruses. Although they do not directly affect humans and animals, these viruses can modulate the transmission of pathogenic viruses to vertebrates [41,42]. The growth of tourism and trade has also led to intensive exchanges of viral pathogens and their vectors in different geographic regions. Together with the rapid growth of large cities in tropical countries, these are the basis for outbreaks or/and epidemics of mosquito-borne infections among animals and humans, creating environments that can maintain the transmission of zoonotic infections [43]. In addition, viruses have extraordinary evolutionary potential for generating new pathogenic isolates that can cause severe diseases in humans and/or animals. 

The south of the Western Siberian Plain is characterized by a continental climate, with short, warm summers and long winters; uniform humidity; and rather abrupt changes in all weather components over relatively short periods of time [44]. This region has experienced characteristic negative mean annual temperatures during the last century, with the maximum variation in the mean annual temperature being 3.6 °C over the observation period. The activity season for different species of mosquitoes begins when the ambient temperature rises above 0 °C (early May) and ends in late August or early September, depending on the year. The maximum duration of the period of mosquito activity is approximately four or five months. Seventeen species of mosquitoes were previously found in the forest–steppe and steppe zones of the region [45]. The composition of mosquito species from different foci can drastically vary. For example, the concentrations of *Cq. richardii* can vary from 1.7 to 99.5%, with this species usually dominating in the main forest–steppe and steppe landscapes of the rural part of the Novosibirsk region. 

In this study, we used a metagenomic-sequencing method to identify the viromes in seven mosquito species collected in the vicinity of Novosibirsk. The metagenomic approach was used to identify the viral diversity in randomly collected mosquitoes. We identified 30 coding-complete viral genomes and viral-like partial sequences of capsid proteins and/or RdRps from the mosquitoes (Table 1). These sequences were classified as putative members of the orders *Tymovirales* and *Picornavirales*; the families *Partitiviridae*, *Totiviridae*, *Tombusviridae*, *Iflaviridae*, *Nodaviridae*, *Permutotetraviridae*, and *Solemoviridae*; and four unclassified RNA-viruses. The previously described Partitivirus-like 1, Hammarskog tombus-like virus, Hammarskog picorna-like virus, Lymantria dispar iflavirus 1, Wenzhou noda-like virus 6, Mayapan virus, Sanxia permutotetra-like virus 1, and Chaq virus-like 1 were identified as practically complete genomes with levels of identity of 82–99% with the *Cq. richardii* mosquito. These viruses were found earlier in Liberia (west Africa), Sweden (north Europe), the USA (America), China (Asia), and Mexico (Central America). These findings allowed us to hypothesize that these viruses may be widely distributed on a global scale. 

Some novel putative viruses and viral sequences had prototype viral sequences with identity levels from 31% to 79%, with these prototypes also being found in invertebrates from almost all continents. Some of them were associated with different species of mosquitoes. In our study, the majority of the novel viruses were associated with the *Cq. richardii* mosquito, a species that is widespread in the south of western Siberia [45]. The role of this mosquito species in the spread of human viral infections in Siberia has been virtually uninvestigated, suggesting that our knowledge concerning mosquito-associated viruses in north Eurasia is very limited and requires further study.

## 5. Conclusions

We identified novel and known viral genomes and viral-like partial sequences in mosquitoes collected in the Novosibirsk region of western Siberia. They were classified as novel putative viruses via a bioinformatics analysis of partial sequences of capsid proteins and RdRps or whole polyproteins (genomes) within the orders *Tymovirales* and *Picornavirales*; the families *Partitiviridae*, *Totiviridae*, *Tombusviridae*, *Iflaviridae*, *Nodaviridae*, *Permutotetraviridae*, and *Solemoviridae*; and four unclassified RNA viruses. We believe that these virus identifications can enhance our understanding of the transmission of RNA viruses by mosquitoes in north Asia. We hope that the discovery and observation of these mosquito-borne viruses can help prevent future outbreaks of viral infections in the region under study.

## Figures and Tables

**Figure 1 viruses-15-01896-f001:**
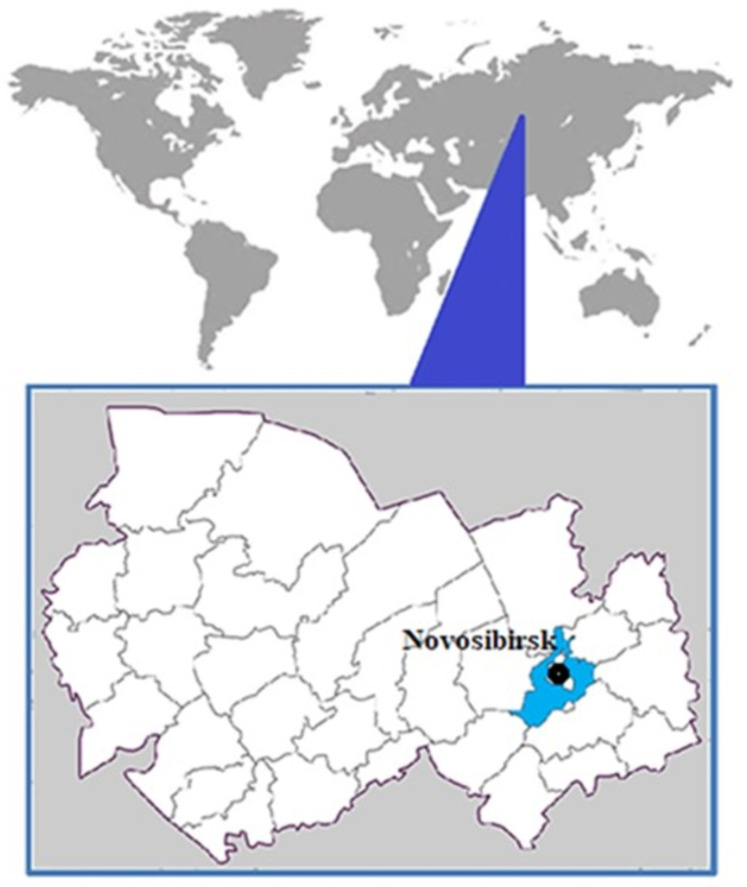
The location from which mosquitoes were collected for this study, with the upper picture showing the world and the lower picture showing the Novosibirsk region.

**Figure 2 viruses-15-01896-f002:**
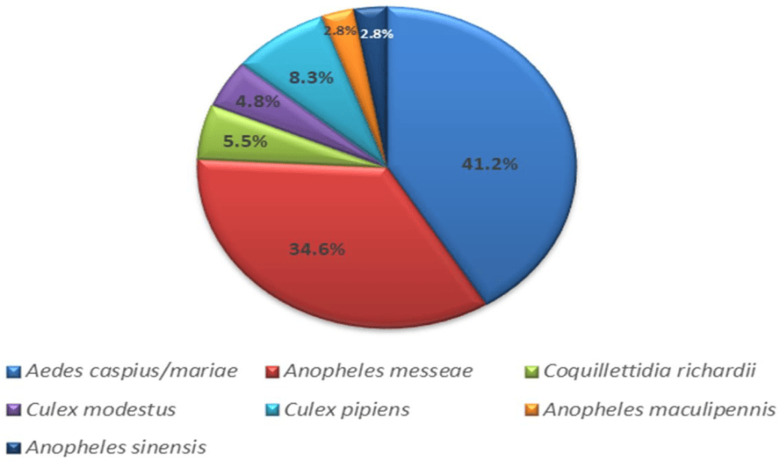
Histogram of mosquito species collected in the southern part of western Siberia (Novosibirsk region).

**Figure 3 viruses-15-01896-f003:**
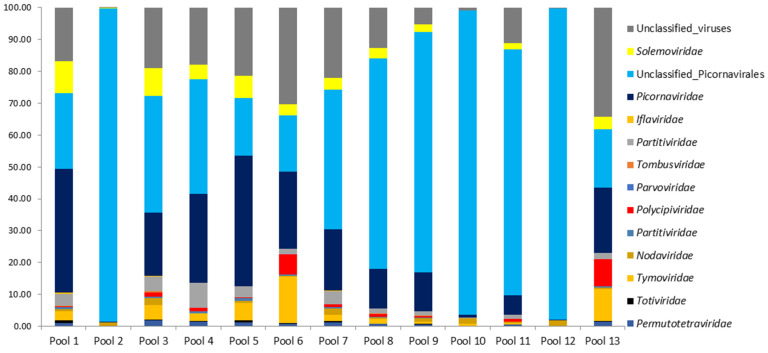
Annotation of the taxonomy for the viral reads in the different pools of the *Cq. richardii* mosquito. Proportions for classified and unclassified sequences are in %.

**Figure 4 viruses-15-01896-f004:**
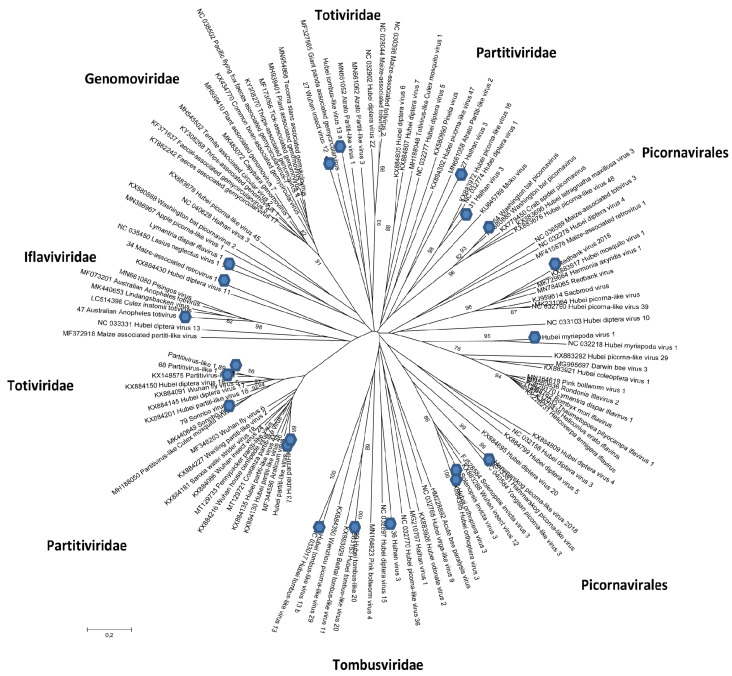
Schematic phylogenetic tree for RNA viruses belonging to different families found in mosquitoes collected in western Siberia in 2017–2018. Maximum likelihood phylogenetic tree based on the amino acid sequences of RdRps. Branch numbers indicate bootstrap support in percentage out of 1000 replicates. The sequences from this study are marked with circles. *Tymovirales* (MW251314; MW251315); *Partitiviridae* (MW251327; MW389552; MW389553; MW389554; MW389555; MW251328; MW251329; MW251330; MW251325; MW389559; MW389560; MW389561); *Totiviridae* (MW251336); *Tombusviridae* (MW251332; MW251324); *Picornavirales* (MT753151; MW251320; MW251321; MW251322; MW251316; MW251317; MW251323; MW251313); *Iflaviridae* (MW251319; MT753152; MT753153; MW389556; MW389557); *Permutotetraviridae* (MT753154; MW389558).

**Table 1 viruses-15-01896-t001:** List of putative and novel mosquito-associated viruses detected in western Siberia.

	Name of Virus	Viral Prototype (Identity, %)	Accession Number, GenBank	Genome (Fragment) Size (bp)	Coverage by NGS, Times
	*Tymovirales* (positive ssRNA)				
1.	Inya insect-associated virus 1	Insect-associated tymovirus 1/Andean potato mild mosaic virus (near 60%)	MW251314	6568 (polyprotein)	155
2.	Inya insect-associated virus 2	Insect-associated tymovirus 1/Andean potato mild mosaic virus (near 60%)	MW251315	6566 (polyprotein)	105
	*Partitiviridae* (dsRNA)				
3.	**Partitivirus-like 1**	**Partitivirus-like 1 (89%)**	**MW251327**	**1749 (RdRp)**	**40**
4.	Krahall insect-associated virus 1/01	Atrato partiti-like virus 2 (63%)	MW389552	1490 (capsid)	179
	Krahall insect-associated virus 1/02	Atrato partiti-like virus 2 (61%)	MW389553	1512 (capsid)	71
	Krahall insect-associated virus 1/03	Atrato partiti-like virus 2 (61%)	MW389554	1488 (capsid)	35
5.	Krahall insect-associated virus 2	Atrato partiti-like virus 2 (60%)	MW389555	1531 (capsid)	79
6.	Talaya insect virus 2/01	Partitivirus-like 1 (68%)	MW251328	1426 (RdRp)	51
	Talaya insect virus 2/02	Partitivirus-like 1 (71%)	MW251329	1489 (RdRp)	30
	Talaya insect virus 2/03	Partitivirus-like 1 (68%)	MW251330	1426 (RdRp)	20
7.	TarBrook virus	Sonnbo virus (79%)	MW251325	1742 (hypot. protein)	75
8.	Zeyabrook partiti-like_virus 1/01	Beihai partiti-like virus 2 (38%)	MW389559	1597 (capsid)	99
	Zeyabrook partiti-like_virus 1/02	Beihai partiti-like virus 2 (38%)	MW389560	1597 (capsid)	31
9.	Zeyabrook partiti-like_virus 2	Beihai partiti-like virus 2 (42%)	MW389561	1458 (capsid)	32
*Totiviridae* (monopartite, dsRNA)					
10.	Zyryana toti-like virus 2	Fitzroy Crossing toti-like virus 1 (51%)	MW251336	>5863 (RdRp, capsid)	24
*Tombusviridae* (positive, ssRNA)					
11.	**Hammarskog tombus-like virus**	**Hammarskog tombus-like virus (90)**	**MW251332**	**4317 (polyprotein)**	**10**
12.	Oyosh tombus-like virus	Hubei tombus-like virus 13 (62%)	MW251324	>5640 (polyprotein)	19
*Picornavirales* (positive, ssRNA)					
13.	**Hammarskog picorna-like virus**	**Hammarskog picorna-like virus (98%)**	**MT753151**	**11506 (polyprotein)**	**6952**
14.	Miltyush picorna-like virus 1/01	Halhan virus 3 (31%)	MW251320	9329 (polyprotein)	31
	Miltyush picorna-like virus 1/02	Halhan virus 3 (31%)	MW251321	9329 (polyprotein)	19
15.	Miltyush picorna-like virus 2	Halhan virus 3 (36%)	MW251322	10,077 (polyprotein)	49
16.	Isses picorna-like virus 1	Washington bat picornavirus (64%)	MW251316	8992 (polyprotein)	537
	Isses picorna-like virus 2	Washington bat picornavirus (64%)	MW251317	8992 (polyprotein)	134
17.	Ora Rivulet insect-associated polycipivirus	Polycipiviridae sp (34%)	MW251323	10,879 (polyprotein)	62
18.	Icha Creek insect virus	Solenopsis invicta virus 3 (32%)	MW251313	10,272 (polyprotein)	24
*Iflaviridae* (positive, ssRNA)					
19.	**Lymantria dispar iflavirus 1**	**Lymantria dispar iflavirus 1 (99%)**	**MT753155**	**9996 (polyprotein)**	**347**
*Nodaviridae* (Bi-partite positive-sense, ssRNA)					
20.	**Wenzhou noda-like virus 6**	**Wenzhou noda-like virus 6 (87%)**	**MW251319**	**3098 (RdRp)**	**159**
21.	**Mayapan virus 1/1**	**Mayapan virus (90%)**	**MT753152**	**3024 (RdRp)**	**292**
	Mayapan virus 1/2	Mayapan virus (88%)	MT753153	1378 (capsid)	181
22.	Mayzas noda-like virus RNA 2	*Mayapan virus segment RNA2 (43%)*	MW389556	1239 (capsid)	68
23.	Uzakla insect-associated virus	Mosinovirus (36%)	MW389557	2493 (capsid)	46
*Permutotetraviridae* (dsRNA)					
24.	**Sanxia permutotetra-like virus 1**	**Sanxia permutotetra-like virus 1 (93%)**	**MT753154**	**4670 (polyprotein)**	**378**
25.	Uzakla mosquito-associated permutotetra-like virus	Vespa velutina permutotetra-like virus 2 (53%)	MW389558	1749 (capsid)	207
Other viruses and viral sequences					
26.	**Chaq virus-like 1/1**	**Chaq virus-like 1 (82%)**	**MW251333**	**1495 (hypot. protein)**	**451**
	Chaq virus-like 1/2	Chaq virus-like 1 (82%)	MW251334	1492 (hypot. protein)	300
27.	Tartas insect associate virus	Atrato Sobemo-like virus 6 (61%)	MW251326	3259 (polyprotein)	464
28.	ZeyaBrook chaq-like_virus 2	Chaq virus-like 3 (47%)	MW251335	1409 (hypot. protein)	220
29.	Kamenka insect-associated virus	Hubei levi-like virus 3 (38%)	MW251318	>3296 (polyprotein)	17
30.	Uzakla insect virus	Hubei myriapoda virus 1 (33%)	MW251331	9614 (polyprotein)	46

Note: Bold letters indicate the sequences with an identity level of 80% or more.

## Data Availability

The data presented in this study are available in the article.

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
