# Peer review of "The Viromes of Mosquitoes from the Natural Landscapes of Western Siberia"

_viruses, 2023, doi:10.3390/v15091896_

Round 1

Reviewer 1 Report

The current work of Ternovoi et al investigates the virome of mosquitoes from western Siberia. The recent pandemic showed how critical is to study the virus host range and transmission thus, I found this study useful which showed the potential of mosquitoes to transmit numerous viruses. This study complements and expands the study done by a group in Sweden on mosquito virome (Öhlund et al., 2019). Given the importance of this study, some gaps are there which remain to be filled before publication that would increase the impact of the work.

Major comments:

- lines 89 to 90 and Fig 4: The methodology for the phylogenetic tree is not well described. Is it a maximum likelihood tree? What model was used to do tree? After aligning the sequences need to be trimmed before running the phylogenetic tree, it was not mentioned. The current Fig 4 needs to be improved. I would recommend using the same sequences reported in Öhlund et al., 2019 and check if the new sequences are making a strong clade with the reported ones. The authors should also report or indicate the nodes with more than 70% confidence. 

- Figure 3 legend is confusing. Is it for all mosquito species or for richardii mosquito? line 116 says for all mosquitos.

- There is an opportunity to discuss the results more which helps the reader to understand the significance of this study and relevant citations should be added. Section 3.2.4 Tombusviridae should be discussed more either here or in the discussion section. Tombusviridae consists of plant-infecting viruses but is frequently found in insects. There is a paper that says tombusvirus may have originated from insect-infecting viruses and this can be supported by the data presented in this paper. Moreover, citation number 27, is not a paper on plant virus thus, proper citations need to be added that discusses viruses from the Tombusviridae family. Some of them are: https://doi.org/10.3390/v13040646; doi: 10.1093/ve/veab030; https://doi.org/10.3390/v15030638

- I would also request to highlight the findings here in contrast to the findings from Öhlund et al., 2019. The new viral sequences that are closely related to the reported ones from Öhlund et al., 2019 should be compared, and if there are non-synonymous mutations in RdRp or capsid protein those should be reported as it may have a role in adaptation to a new environment.

I would request authors to do a thorough checkup for the grammar and scientific language. e.g., in line 22, instead of "main part of the", it would be "interestingly"

Author Response

Comments for reviewer 1

Thank you for considering our manuscript: ID viruses-2530126  «Virome of Mosquitoes from Natural Landscapes of the Western Siberia”  for publication in Viruses.

We thank you for valuable suggestions that allowed us to make the manuscript (MS) more convincing and understandable. We accepted your suggestion and made corresponding changes in the MS.

Below please find our detailed responses to your questions and comments.

Yours sincerely,  

Valery B. Loktev

You wrote: 

- lines 89 to 90 and Fig 4: The methodology for the phylogenetic tree is not well described. Is it a maximum likelihood tree? What model was used to do tree? After aligning the sequences need to be trimmed before running the phylogenetic tree, it was not mentioned. The current Fig 4 needs to be improved. I would recommend using the same sequences reported in Öhlund et al., 2019 and check if the new sequences are making a strong clade with the reported ones. The authors should also report or indicate the nodes with more than 70% confidence. 

The preliminary draft of Figure 4 took up more than 6 typewritten pages and we tried to find a more concise and informative form for this figure.  We hope that a simplified circular dendograme is more representative and illustrative.     

In addition, the note for Fig4 has been modified by adding a more details.  Also, references to figures S1-S7 from the Supplemental section are available into the MS.  This section contains an additional methodology details, indicate the confidence support to main nodes, the names of the sequences used for phylogenetic analysis, and the structural features of the sequences and genome for this study. Also, we tried to improve the quality of Figure 4, making it with a resolution of 1200 dpi. The reference Öhlund et al., 2019 (doi: 10.3390/v11111027) with the sequences are included and discussed to MS.  

- Figure 3 legend is confusing. Is it for all mosquito species or for richardii mosquito? line 116 says for all mosquitos. 

Of course, the MS has been modified

- There is an opportunity to discuss the results more which helps the reader to understand the significance of this study and relevant citations should be added. Section 3.2.4 Tombusviridae should be discussed more either here or in the discussion section. Tombusviridae consists of plant-infecting viruses but is frequently found in insects. There is a paper that says tombusvirus may have originated from insect-infecting viruses and this can be supported by the data presented in this paper. Moreover, citation number 27, is not a paper on plant virus thus, proper citations need to be added that discusses viruses from the Tombusviridae family. Some of them are: https://doi.org/10.3390/v13040646; doi: 10.1093/ve/veab030; https://doi.org/10.3390/v15030638

This section of the manuscript has been modified and these references have been introduced into the MS.

- I would also request to highlight the findings here in contrast to the findings from Öhlund et al., 2019. The new viral sequences that are closely related to the reported ones from Öhlund et al., 2019 should be compared, and if there are non-synonymous mutations in RdRp or capsid protein those should be reported as it may have a role in adaptation to a new environment.

You are absolutely right. The ratio of non-synonymous and synonymous substitutions (dN/dS) is very interesting and may indicate the direction of changes in the genome.  Our small experience with the use of this indicator (Molecular evolution of the tick-borne encephalitis and Powassan viruses. Mol Biol (Mosk). 2012 Jan-Feb;46(1):82-92)  suggests  that usually  needs representative set of isolates to obtain interesting results.

In this case, the level of homology was 98.4% for amino acid sequence for Hammarskog picorna-like virus isolate Koltsovo/HPLV/2019 (MT753151) and Hammarskog picorna-like virus (MN513381). We found 150 non-synonymous substitutions and 640 synonymous substitutions, dN/dS ratio - 0,23 that is typical for RNA viruses.

Comments on the Quality of English Language

I would request authors to do a thorough checkup for the grammar and scientific language. e.g., in line 22, instead of "main part of the", it would be "interestingly"

The MS has been modified and we also used MDPI Language Editing Services for editing all text of MS (certificate enclosed).

Thank you for your remarks to MS again. Now, revised MS looks really better.

Reviewer 2 Report

The authors conducted a metagenomic survey of RNA viruses in mosquitoes collected in Western Siberia. They found several novel viral RNA sequences of some viral orders, families, and unclassified RNA viruses. There are many similar studies using NGS, but most study sites are tropical and subtropical areas. The unique point of this study is that the authors focused on their study site in Dfb of climate classification, where mosquito-borne infection is uncommon.

Major revision

The manuscript mainly focuses on viruses but not mosquito features and climates, such as mosquito phylogenetical classification and evolution. If the authors include the above information, the manuscript will fit more into the Special Issue's concept, Ecology.

Minor revision

Page 2, line 63-65

Did authors read 16S rRNA and COI sequence before or after mosquito pooling? Was it by Sanger sequencing? Is there no contamination of different species in one pool?

Figure 4

The label of each virus name is unreadable.

Author Response

Comments for reviewer 2

Thank you for considering our manuscript: ID viruses-2530126  «Virome of Mosquitoes from Natural Landscapes of the Western Siberia”   for  publication in Viruses.

We thank you for valuable suggestions that allowed us to make the manuscript more convincing and understandable. We accepted your suggestion and made corresponding changes in the manuscript.

Below please find our detailed responses to your questions and comments.

Yours sincerely,  

Valery B. Loktev

You wrote:

Major revision

The manuscript mainly focuses on viruses but not mosquito features and climates, such as mosquito phylogenetical classification and evolution. If the authors include the above information, the manuscript will fit more into the Special Issue's concept, Ecology.

 The MS has been revised as recommended by reviewers. We also used MDPI Language Editing Services for editing all text of MS (certificate enclosed).

Minor revision

Page 2, line 63-65

Did authors read 16S rRNA and COI sequence before or after mosquito pooling? Was it by Sanger sequencing? Is there no contamination of different species in one pool?

Yes of course, by Sanger and during pooling, in parallel. The additional details have been included to MS 

Figure 4 

The preliminary draft of Figure 4 took up more than 6 typewritten pages and we tried to find a more concise and informative form for this figure.  We hope that a simplified circular dendograme is more representative and illustrative. In addition, the note for Fig4 has been modified by adding a more details.  Also, references to figures S1-S7 from the Supplemental section are available into the MS.  These contains additional information for methodology details, indicate the confidence support to main nodes, the names of the sequences used for phylogenetic analysis, and the structural features of the sequences and genome for this study. Also, we tried to improve the quality of Figure 4, making it with a resolution of 1200 dpi.  

Thank you for your remarks to MS again. Now, revised MS looks really better.

Round 2

Reviewer 1 Report

I would like to thank the authors for taking the time to improve the manuscript. The phylogenetic tree has also improved however, it is still challenging to read due to the overlapping of names. I would suggest collapsing some of the nodes from each viral family. This will increase the clarity of the image. In material and methods, it can be mentioned what viral sequences were used and present in the collapsed nodes. line 228 can be written as "and they are sometimes associated with subviral RNA molecule capable of replicating itself".

English is readable and understandable. 

Author Response

August 28, 2023

Comments for reviewer

Thank you for considering our manuscript: ID viruses-2530126  «Virome of Mosquitoes from Natural Landscapes of the Western Siberia”  for publication in Viruses.

We thank you for valuable suggestions that allowed us to make the manuscript (MS) more convincing and understandable. We accepted your suggestion and made corresponding changes in the MS. The MS was editing by MDPI proofediting service, the MDPI certificate enclosed.

Below please find our detailed responses to your questions and comments.

Yours sincerely,  

Valery B. Loktev

You wrote: 

The phylogenetic tree has also improved however, it is still challenging to read due to the overlapping of names. I would suggest collapsing some of the nodes from each viral family. This will increase the clarity of the image. In material and methods, it can be mentioned what viral sequences were used and present in the collapsed nodes. line 228 can be written as "and they are sometimes associated with subviral RNA molecule capable of replicating itself".

Yes of course,  we completely revised figures 4 of MS and MS has been modified. Also, all other modifications are marked by yellow
